# Is the Severity of Cervical Foraminal Stenosis Related to the Severity and Sidedness of Symptoms?

**DOI:** 10.3390/healthcare9121743

**Published:** 2021-12-17

**Authors:** Han-Dong Lee, Chang-Hoon Jeon, Nam-Su Chung, Ha-Seung Yoon, Hee-Woong Chung

**Affiliations:** Department of Orthopaedic Surgery, Ajou University School of Medicine, Suwon 16499, Korea; handonglee@aumc.ac.kr (H.-D.L.); ajouosspine@gmail.com (C.-H.J.); ajouosns@gmail.com (N.-S.C.); yoonhaseung@naver.com (H.-S.Y.)

**Keywords:** cervical foraminal stenosis, neck pain, arm pain, neck disability index, grade, severity, sidedness

## Abstract

(1) Background: Cervical foraminal stenosis (CFS) is a common cause of axial neck and arm pain. The aim of this study was to determine the relationship between the severity of CFS and clinical symptoms in terms of severity and sidedness. (2) Methods: We retrospectively reviewed 75 consecutive patients with degenerative CFS. We graded 900 foramina from C3–4 to T1–2 using Park’s grading system. We collected visual analogue scale (VAS) and neck disability index (NDI) values from the neck and both arms. We analyzed the relationships with CFS grades and total number. We defined four types of left/right dominance of CFS (none, left-dominant, right-dominant, and both) by comparing left and right sides using total counts and maximal grade of CFS, respectively. We compared arm pain sidedness (no arm pain, left-only, right only, and bilateral) among different left and right CFS dominance types. (3) Results: Mean neck and left and right arm VAS scores were 4.4 ± 2.5, 4.9 ± 1.6, and 4.6 ± 2.0, respectively. The mean total NDI was 16.0 ± 8.0. The CFS grade at C3–4 and total count were correlated with neck VAS. Arm VAS was also correlated with CFS grade and total counts. Total NDI score was not correlated with radiological parameters. The presence and sidedness of arm pain were significantly different between left and right CFS dominance groups divided by total count of grade 2 and 3 CFS. (4) Conclusions: The CFS grade and total count were correlated with neck and arm VAS. Arm pain occurred more frequently on the side with more grade 2 and 3 CFS.

## 1. Introduction

Cervical foraminal stenosis (CFS) is a common cause of cervical radiculopathy and is associated with axial neck and arm pain [1]. A previous study indicated that about 10–25% of the adult population have CFS [2]. Acute, mild symptoms of CFS respond well to conservative treatment, such as pain medication [3,4], epidural injection [5], exercise [6], and manipulation [7,8]. Surgical treatment is recommended for patients with persistent or severe radicular symptoms that are unresponsive to conservative treatment [9,10,11,12]. For successful injection or surgical treatment, it is important to determine the level responsible for the symptoms. However, there are no universally accepted diagnostic criteria for CFS [13].

Thorough patient history taking and physical examination are helpful in diagnosis of the level responsible for symptoms [1]. However, it is sometimes difficult because radicular pain does not always follow commonly used dermatomal maps [14]. It is more difficult to diagnose the responsible level in patients with multilevel CFS.

Magnetic resonance imaging (MRI) is an important modality for evaluating CFS, providing detailed visualization of the soft tissue, such as intervertebral discs and neural structures. Recently, T2 reconstructed oblique MRI was shown to provide better information about the cervical intervertebral foramen, which is located anterior to the vertebral canal at an angle of 45° relative to the coronal plane [15,16,17]. Park et al. introduced a grading system for CFS using oblique sagittal MRI. This method is reliable with good intra- and interobserver agreement and is correlated with the presence of neurological manifestations [18,19,20].

However, there have been no reports regarding the relationship between CFS grade and the severity and sidedness of clinical symptoms. Therefore, this study was performed to determine this relationship.

## 2. Materials and Methods

### 2.1. Patients

We retrospectively reviewed 254 consecutive patients referred to a tertiary hospital by general practitioners or medical specialists for neck or arm radiating pain who underwent MRI between April 2013 and June 2019. The inclusion criteria for this study were a diagnosis of CFS (≥grade 1 CFS) due to disk herniation and/or stenosis and age of at least 18 years. Patients with a prior history of cervical spinal surgery (*n* = 5), central canal compression or signs of myelopathy (*n* = 90), inflammatory diseases such as rheumatic arthritis (*n* = 4), tumors or neural cysts (*n* = 9), acute trauma (*n* = 37), congenital anomalies (*n* = 2), infection (*n* = 0), brain disease (*n* = 2), shoulder disease (*n* = 2), peripheral nerve disease in the upper extremities (*n* = 8), incomplete clinical score (*n* = 8), MRI more than 3 months before clinical scoring (*n* = 3), no reconstructed oblique MRI (*n* = 5), and no foraminal stenosis (*n* = 4) were excluded. In the end, 75 patients were included in the analysis. All patients underwent clinical scoring of the neck and both arms using visual analogue scales (VAS) and the neck disability index (NDI) at the first visit to our outpatient clinic.

### 2.2. Reconstructed Oblique Sagittal MRI and CFS Grading

All MRIs were obtained using a 1.5 T magnet in a single institute. Left and right oblique T2-weighted images were created by reformatting the images in a plane perpendicular to the long axis of the neural foramina, 45° from the sagittal and coronal planes. The images included the lateral edge of the foramen, the isthmus of the foramen, and the medial margin of each pedicle with a slice thickness of 3 mm and interslice gap of 0 mm.

One experienced spinal surgeon, who was blinded to the clinical information of patients, graded 900 neural foramina from C3–4 to T1–2 using Park’s method (Figure 1) [19]: Grade 0 = no stenosis; Grade 1 = mild stenosis with perineural fat obliteration < 50% of nerve root circumference; Grade 2 = moderate stenosis with perineural fat obliteration > 50% of nerve root circumference; Grade 3 = severe stenosis, collapsed nerve root, and morphological changes of the nerve root.

### 2.3. Left/right Dominance of CFS and Arm Pain Sidedness

We defined four types of left/right CFS dominance (none, left-dominant, right-dominant, and both) using the total number of foramina with CFS grade ≥ 1, 2, and 3 and the maximal grade of CFS, respectively. CFS grade ≥ 1-type left/right dominance was “left” if the total number of foramina with CFS grade ≥ 1 on the left side was greater than the number on the right side in a patient, and “right” if vice versa. “Both” indicated equal numbers on both sides. CFS grade ≥ 2 and 3 types were defined in the same way. The maximal grade of CFS type left/right dominance was defined by comparing the highest grade of left and right sides of CFS.

Arm pain sidedness was divided into none, left only, right only, or bilateral: none = VAS of 0 for both arms; left only = right arm VAS of 0 and left arm VAS > 0; right only = right arm VAS > 0 and right arm VAS of 0; bilateral = VAS > 0 for both left and right arms.

### 2.4. Statistical Analyses

Pearson’s correlation analysis was adopted to determine relationships between all parameters. The relationships between CFS grades at all 12 foramina (left and right foramina from C3–4 to T1–2) and neck and bilateral arm VAS and NDI scores were analyzed. Relationships between maximal grades of the left, right, and both sides and clinical parameters were analyzed as well as the relationships between the total number of foramina with CFS grade ≥ 1, 2, and 3 on the left, right, and both sides and clinical parameters. We compared arm pain sidedness among four different CFS left/right dominance groups created by CFS grade ≥ 1, 2, and 3 and the maximal grade of CFS, respectively, using the Chi-squared test. Post hoc analysis using the Chi-squared test was performed to evaluate the significance of differences between the groups. Statistical analyses were performed using SPSS statistical software (SPSS 20.0; IBM, Armonk, NY, USA). In all analyses, *p* < 0.05 was taken to indicate statistical significance.

## 3. Results

### 3.1. Demographic Data

The study population had a mean age of 51.0 ± 13.1 years (range: 19–82), and consisted of 30 men and 45 women. Mean neck and left and right arm VAS scores were 4.4 ± 2.6, 4.9 ± 1.6, and 4.6 ± 2.0, respectively. Mean total NDI was 16.0 ± 7.6. NDI percentage was 32.8% ± 8.6%. Only three patients had one-level CFS. Mean total counts of foramina with CFS grades 1, 2, and 3 were 3.4 ± 1.9, 0.5 ± 0.9, and 1.9 ± 1.9, respectively. The distribution of CFS according to cervical level is shown in Table 1. The most frequently involved level of CFS was C5–6 on both sides. Clinical parameters demonstrated no relationships with age or gender. CFS grades were positively correlated with age at left C3–4 (r = 0.247, *p* = 0.033) and C4–5 (r = 0.370, *p* = 0.001), and right C4–5 (r = 0.351, *p* = 0.002) and C6–7 (r = 0.261, *p* = 0.024). Gender was not related to CFS grade.

### 3.2. CFS Grades and Severity of Clinical Outcome

Neck VAS score was correlated with left and right C3–4 CFS grades (r = 0.244, *p* = 0.035; r = 0.253, *p* = 0.029, respectively). Neck VAS was also correlated with the total count of both foramina with CFS grade ≥ 1 (r = 0.230, *p* = 0.048). Left arm VAS was not correlated with CFS grade at any level. The total count of left foramina with CFS grade ≥ 1 was correlated with left arm VAS score (r = 0.258, *p* = 0.026). Right arm VAS score was correlated with right C5–6 CFS grade (r = 0.231, *p* = 0.046). It was not correlated with the total CFS count. Total NDI score was not correlated with CFS grade or total CFS count. Maximal CFS grades on the left side, right side, and both sides were not correlated with any clinical parameters.

### 3.3. CFS Grades and Sidedness of Arm Pain

Overall, grade 1, 2, and 3 CFS on the left side were seen in 138 (27.9%), 23 (4.6%), and 70 (14.1%) cases, respectively. The corresponding numbers on the right side were 127 (25.7%), 16 (3.2%), and 78 (15.8%), respectively. Left/right dominance of CFS, categorized using the total number of foramina with CFS ≥ grades 1, 2, and 3, was 0/25/22/28 (none/left-dominant/right-dominant/both), 18/17/21/19, and 24/13/22/16, respectively. Left/right dominance of CFS categorized using the maximal grade of CFS was 0/10/13/52. Arm pain sidedness was 21/22/24/8 (none/left only/right only/bilateral). There were no significant differences in arm pain sidedness between left/right dominance CFS groups divided by the total number of foramina with CFS ≥ grade 1 and maximal CFS grade. Arm pain sidedness was significantly different between left/right dominance groups categorized by grade 2 and 3 CFS (*p* = 0.023 and 0.001, respectively). In subgroup analysis, arm pain sidedness was significantly different among left/right dominance groups categorized only by grade 3 CFS (Table 2, Table 3, Table 4 and Table 5).

## 4. Discussion

In the present study, in degenerative CFS patients without central stenosis, CFS grade and total count were shown to be correlated with neck and arm VAS. Neck VAS was correlated with CFS grade at the C3–4 level, right arm VAS was correlated only with CFS grade at the right C5–6 level, and left arm VAS was correlated with total count of CFS at the left neural foramina. Total NDI score was not correlated with CFS grade or total count. The maximal grade of CFS was not correlated with any of the clinical parameters examined. The presence and sidedness of arm pain were significantly related to grade 2 and 3 CFS.

While many groups have attempted to determine the relationships between radiological findings regarding lumbar foraminal stenosis and clinical symptoms in order to confirm the symptom level [21,22,23,24], there have been few studies regarding cervical foraminal stenosis. Several grading systems for CFS have been reported [19,25]. However, only Park’s method has been demonstrated as reliable; practical, without the need for quantitative measurement; and related to clinical manifestations [18]. Park et al. analyzed 166 patients to determine the relationship between their CFS grading of C4–5, C5–6, and C6–7 and corresponding neurological symptoms. They defined positive neurological manifestations of corresponding cervical neural foraminal stenosis as more than one positive neurological clinical manifestation (paresthesia, extremity weakness, numbness, and funicular or radicular pain) combined with more than one positive neurological sign (positive Lhermitte sign, Spurling sign, decreased deep tendon reflex response, and positive denervation sign on electromyography). They found that grades 2 and 3 were associated with positive neurological manifestations. One weakness of the study of Park et al. was that clinical manifestations were too simple and not quantitative. We used patient-reported VAS scores and NDI score for the neck and both arms instead of assessment by physicians. In the present study, performed mainly in patients with multilevel CFS, maximal grade of CFS was not correlated with any of the clinical symptoms. We found that arm pain occurred more frequently on the side with the larger count of grade 2 and 3 CFS in this multilevel dominant CFS population. Only grade 3 CFS was related to arm pain sidedness in subgroup analysis. More perineural fat obliteration (in particular, >50% of nerve root circumference) is related to poorer clinical features in the lumbar spine [22,23], and findings in the cervical spine were similar. The results of the present study and previous studies suggest that grade 2 and 3 CFS, especially grade 3 CFS in the side with arm pain, may be related to clinical symptoms and may represent a target for treatment.

We also found that CFS grades at C3–4 and total count were correlated with neck VAS score. Neck VAS score was correlated with CFS grade on both sides at the C3–4 level. This was consistent with two previous studies that showed a relationship between C4 radiculopathy and neck pain [26,27]. Even though these two studies showed that symptoms in different neck regions (suboccipital vs. base of posterior neck) can be caused by radiculopathy due to C3–4 level pathology, both suggested that neck pain is one of the symptoms of C4 radiculopathy. C3–4 level CFS in combination with neck pain could potentially be a treatment target. Total count of CFS ≥ grade 1 was also correlated with neck VAS score. These findings suggest that all CFS grades may contribute to neck pain. However, the total count of CFS may have been a confounder of neck pain in this study because CFS is usually combined with cervical spinal pathology related to neck pain, such as disc degeneration and facet joint arthritis [28].

In the present study, NDI was not related to any radiological parameters. This was contrary to the findings of previous studies indicating a significant improvement in NDI after foraminal decompression [29]. However, findings similar to those of the present study were reported in a previous study of lumbar spinal stenosis, in which Oswestry Disability Index (ODI) score was not correlated with the radiological severity of lumbar spinal stenosis [21]. They assumed that the discrepancy between ODI and radiological findings may have been caused by the wide variability in lumbar dimension (in patients with a severe grade of stenosis and no symptoms) or dynamic factors (in patients with a mild degree of stenosis and severe symptoms). They also suggested that other factors, such as mood, have a more important relationship with ODI. We also assumed that other factors, such as dynamic compression or cervical alignment, may have a greater influence on NDI than the severity of CFS.

Our study had a number of limitations. First, this was a retrospective study and, thus, it was possibly subject to bias. To avoid this problem, the MRI observers were blinded to clinical outcomes. Second, the study population was relatively small. The number of cases with CFS at C7–T1 and T1–2 was very limited. This may have been related to the observation that arm VAS score was correlated only with CFS level C5–6 on the right side. Third, we only performed MRI in the supine position. Further dynamic studies may improve the diagnostic accuracy of CFS. However, dynamic studies require specialized equipment, and the results of the present study using supine MRI may be more helpful in general medical situations. Fourth, arm pain in the present study population may not have been of cervical origin. Our experienced spine surgeon performed thorough history-taking and physical examination. Patients with no typical symptoms of CFS were referred to shoulder, elbow, or hand specialists or neurologists. Even though there was a possibility that arm pain may have been caused by upper extremity or peripheral nerve problems, our results may be helpful for the diagnosis and treatment of CFS patients. Finally, this was a cross-sectional study. Further longitudinal studies of treatment according to CFS grade may yield better results regarding confirmation of symptomatic level.

## 5. Conclusions

The CFS grade at C3–4 and total count were correlated with neck VAS. The arm VAS was also correlated with the CFS grade and total counts. Arm pain occurred more frequently on the side with the larger counts of grade 2 and 3 CFS, especially grade 3 CFS. The findings of the present study suggest that C3–4 level CFS in combination with neck pain and grade 2 and 3 CFS on the side with arm pain may be treatment targets. However, the correlations between clinical parameters and radiological parameters were weak and, therefore, degenerative CFS is still a clinicoradiological syndrome. Examination of both clinical symptoms and MRI is important when determining symptomatic level in patients with this diagnosis.

## Figures and Tables

**Figure 1 healthcare-09-01743-f001:**
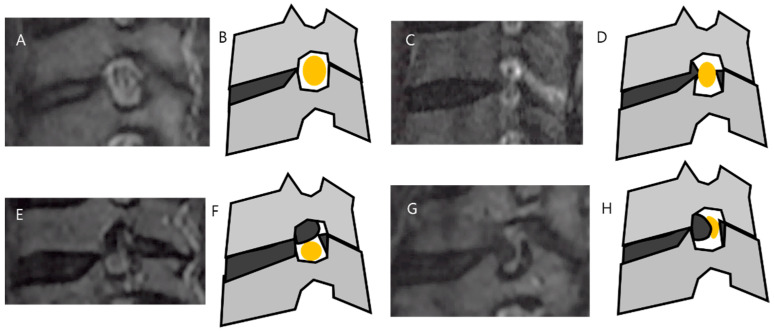
T2 reconstructed oblique sagittal magnetic resonance images and schematic drawing of cervical foraminal stenosis grades 0 (**A**,**B**), 1 (**C**,**D**), 2 (**E**,**F**) and 3 (**G**,**H**). Grade 0 = no stenosis; Grade 1 = mild stenosis with perineural fat obliteration < 50% of nerve root circumference; Grade 2 = moderate stenosis with perineural fat obliteration > 50% of nerve root circumference; Grade 3 = severe stenosis collapsed nerve root and morphological changes of the nerve root.

**Table 1 healthcare-09-01743-t001:** Distribution of foraminal stenosis by vertebral level.

	Park Grade								Total	
	0		1		2		3			
	Left	Right	Left	Right	Left	Right	Left	Right	Left	Right
C3–4	29 (38.7)	32 (42.7)	33 (44.0)	32 (42.7)	3 (4.0)	1 (1.3)	10 (13.3)	10 (13.3)	46 (61.3)	43 (57.3)
C4–5	28 (37.3)	30 (40.0)	34 (45.3)	28 (37.3)	6 (8.0)	5 (6.7)	7 (9.3)	12 (16.0)	47 (62.7)	45 (60.0)
C5–6	12 (16.0)	10 (13.3)	24 (32.0)	31 (41.3)	7 (9.3)	3 (4.0)	32 (42.7)	31 (41.3)	63 (84.0)	65 (86.7)
C6–7	29 (38.7)	26 (34.7)	20 (26.7)	24 (32.0)	5 (6.7)	5 (6.7)	21 (28.0)	20 (26.7)	46 (61.3)	49 (65.3)
C7–T1	50 (66.7)	59 (78.7)	23 (30.7)	10 (13.3)	2 (2.7)	1 (1.3)	0 (0.0)	5 (6.7)	25 (33.3)	16 (21.3)
T1–2	71 (94.7)	72 (96.0)	4 (5.3)	2 (2.7)	0 (0.0)	1 (1.3)	0 (0.0)	0 (0.0)	4 (5.3)	3 (4.0)
Total	219 (48.7)	229 (50.9)	138 (30.7)	127 (28.2)	23 (5.1)	16 (3.6)	70 (15.6)	78 (17.3)	231 (25.7)	211 (23.4)

Data in parentheses are percentages.

**Table 2 healthcare-09-01743-t002:** Distribution of CFS dominance divided by total counts of CFS ≥ grade 2 on left and right sides and arm pain sidedness.

		Arm Pain				Total
		None	Left Only	Right Only	Both	
CFS dominance	None	8	3	3	4	18
	Left	3	10	2	2	17
	Right	5	5	10	1	21
	Both	5	4	9	1	19
Total		21	22	24	8	75

**Table 3 healthcare-09-01743-t003:** Subgroup analysis of CFS dominance divided by total counts of CFS ≥ grade 2 on left and right sides for arm pain sidedness.

		*p*
None	vs. Left	0.076
	vs. Right	0.087
	vs. Both	0.132
Left	vs. Right	0.058
	vs. Both	0.051
Right	vs. Both	0.996

**Table 4 healthcare-09-01743-t004:** Distribution of CFS dominance according to total counts of CFS ≥ grade 3 on left and right sides and arm pain sidedness.

		Arm Pain				Total
		None	Left Only	Right Only	Both	
CFS dominance	None	11	5	3	5	24
	Left	4	8	0	1	13
	Right	3	5	13	1	22
	Both	3	4	8	1	16
Total		21	22	24	8	75

**Table 5 healthcare-09-01743-t005:** Subgroup analysis of CFS dominance divided by total counts of CFS ≥ grade 3 on left and right sides and arm pain sidedness.

		*p*
None	vs. Left	0.073
	vs. Right	0.004 *
	vs. Both	0.039 *
Left	vs. Right	0.006 *
	vs. Both	0.026 *
Right	vs. Both	0.948

* *p* < 0.05.

## Data Availability

The data presented in this study are available on request from the corresponding author.

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
