# Peer review of "Is the Severity of Cervical Foraminal Stenosis Related to the Severity and Sidedness of Symptoms?"

_healthcare, 2021, doi:10.3390/healthcare9121743_

Round 1

Reviewer 1 Report

Good design, performance and analysis.

Author Response

We appreciate your kind review of our manuscript. 

Reviewer 2 Report

The authors have Presented the retrospective review evaluating cervical foraminal stenosis as a cause of worsening neck pain, axial and correlate the degree of cervical foraminal stenosis with the sidedness of the pain.

   they reviewed about 274 patients and following exclusions had 75 patients which the utilized in the study.  the patient is MRIs were evaluated of the cervical spine and T2 oblique views of the cervical foramen were used to grade degree of the stenosis based on Parks methodology.  once the graded the degree and number of levels the correlated it with the sidedness of pain and the severe using the VA AS and ND I.  This then completed a statistical analysis for the appropriate variables and presented that outcomes.

  The study variables and the statistics can be performed, although the results may defer because of the degree of by us which is present in the study and the small number of patients utilized.  The statistical tables are clear and concise, the site a shins are adequate and there is no imaging.

  There also is in the discussion have clearly noted the weakness is of the study and gone to discuss the results which includes a significant correlation between axial neck pain and C3-C4 stenosis and the lack of correlation between sidedness, severity and cervical foraminal stenosis at multiple levels and on the side.  As practitioner as we know that there is many times a lack of correlation between the patient's clinical symptoms and the side of cervical foraminal stenosis which helps us decide which patient's may do better with medical management versus surgical intervention.

  The conclusion is clear.  The study has clinical  relevance to the practicing spine surgical population, although limited by its weaknesses.   the authors may benefit by looking at outcomes following surgery in the future

    minor grammar corrections are needed in the   paper

Author Response

You clearly understood the purpose of our research. Thanks for the good advice.

Grammar was previously reviewed through textcheck.com, but for better quality, it was re-edited at the site recommended by the publisher.

Reviewer 3 Report

Authors have been tried to establish a relationship between cervical foraminal stenosis and clinical symptoms and signs of cervicobrachial syndrome. 

The paper doesn`t offer any new insights into this clinical entity. Methods and results are very hard for understanding. It is obvious that foraminal stenosis at the right side will cause right side pain in the arm, C3/4 stenosis also according to nowadays knowledge cause neck pain???

In general, the paper doesn`t provide any useful advice on how to use described radiological findings for clinical treatement.

This paper needs major revision and another review after that. 

Best regards!

Author Response

In clinical practice, it is often found that not all lesions contribute to symptoms when cervical foramen stenosis is present at multiple levels. Severe stenosis on the right side often does not cause pain in the right arm. Therefore, the purpose of this study was to identify factors related to the location and severity of pain in multi-level cervical foramen stenosis. This study first showed that not only the grade of cervical foramen stenosis, but also the number of segments affected by cervical stenosis is important. In addition, previous studies have shown that cervical foramen stenosis can cause neck pain. And this study reinforces the evidence for such a point.

Please reconsider the intent of our study and the value of its results.

Also, if you can tell us specifically about the parts that require major corrections, we will correct them.

Thank you very much.

Round 2

Reviewer 3 Report

The authors made me change my first opinion, with their comments, about the importance of the topic. 

I still think that the paper is hard to understand in some segments.  

The paper could be helpful for less experienced doctors who are dealing with this pathology.

After one more careful reading, I think that the paper deserves to be published. 

Best regards!